# scRNAseq and High-Throughput Spatial Analysis of Tumor and Normal Microenvironment in Solid Tumors Reveal a Possible Origin of Circulating Tumor Hybrid Cells

**DOI:** 10.3390/cancers16071444

**Published:** 2024-04-08

**Authors:** Abdullah Mahmood Ali, Azra Raza

**Affiliations:** 1Department of Medicine, Division of Hematology/Oncology, Columbia University Irving Medical Center, New York, NY 10032, USA; 2Edward P Evans MDS Center, Herbert Irving Comprehensive Cancer Center, New York, NY 10032, USA

**Keywords:** tumor–macrophage hybrids, circulating hybrid cells, fusion, phagocytosis, CHC, CAML

## Abstract

**Simple Summary:**

This study explored a potentially significant player in the spread of metastatic cancer: “hybrid cells” (HCs) combining features of both epithelial (common in tissues) and immune cells (such as macrophages). HCs were found in more than one type of tumor (THC) and showed a unique transcriptional profile including the activation of functional pathways, potentially explaining their mobility, dissemination into circulation, and metastatic progression. Normal healthy tissue also showed hybrid cells (NHC), but they were rare and could be easily distinguished from THCs based on their gene expression profile. The study also developed a way to identify HCs in large datasets generated at the single-cell level, paving the way for further research on their function and as potential therapeutic targets.

**Abstract:**

Metastatic cancer is a leading cause of death in cancer patients worldwide. While circulating hybrid cells (CHCs) are implicated in metastatic spread, studies documenting their tissue origin remain sparse, with limited candidate approaches using one–two markers. Utilizing high-throughput single-cell and spatial transcriptomics, we identified tumor hybrid cells (THCs) co-expressing epithelial and macrophage markers and expressing a distinct transcriptome. Rarely, normal tissue showed these cells (NHCs), but their transcriptome was easily distinguishable from THCs. THCs with unique transcriptomes were observed in breast and colon cancers, suggesting this to be a generalizable phenomenon across cancer types. This study establishes a framework for HC identification in large datasets, providing compelling evidence for their tissue residence and offering comprehensive transcriptomic characterization. Furthermore, it sheds light on their differential function and identifies pathways that could explain their newly acquired invasive capabilities. THCs should be considered as potential therapeutic targets.

## 1. Introduction

Death from metastatic cancer remains one of the leading causes of mortality from cancers [1]. The process of metastasis involves the tumor cells leaving the primary site and landing and establishing at distant sites [2,3]. Many different theories are proposed to explain the process of metastasis, but a relatively understudied and one of the oldest theories suggests fusion of tumor cells with leukocytes including a monocyte or macrophage results in a hybrid cell (HC) that aids and abets in metastasis [2,3]. In 1908, Otto Aichel suggested that cancerous cells are able to fuse with leukocytes, creating malignant tumor hybrid cells (THCs) [4]. While the concept of THC is old, the studies characterizing THCs from patients are relatively nascent and are limited to a small number of investigators across the globe, mainly due to limitations in THC detection technologies and isolation methods. Recent advances in circulating tumor cells (CTC) isolation and detection technologies identified a new type of cell along with CTC in circulation that is characterized by the presence of a hybrid proteome resembling tumor and immune cell lineages resulting in the resurgence of the concept proposed by Aichel [5,6,7]. These hybrid cells were described using various nomenclatures, including circulating hybrid cells (CHC), tumor macrophage hybrid (TMH), tumor hybrid cells (THC), and cancer associated macrophage-like (CAML, these are generally large and polyploid) cells [5,6,7,8,9,10,11,12,13,14,15,16,17,18,19].

Several studies have shown a significant relation between the presence, size, and/or number of HCs in circulation to tumor stage, progression, and/or survival [9,10,11,13,14,19,20,21,22]. The process of hybrid cell formation between a tumor and an immune cell has been documented in cell-culture and animal models [9,19]. While the source of these cells in circulation is believed to be the primary and/or metastatic tumor, a few studies have demonstrated their presence, mainly using immunohistochemistry, in the primary tumor using a limited number of markers and candidate approaches [23,24,25,26,27,28]. Due to a lack of markers and a consensus concerning the definition of these cells, several different proteins, including CD14, CD45, CD68, and/or CD163, to identify immune lineage and EPCAM and/or pancytokeratins, including CK8, CK9, and CK19, to identify epithelial lineage were used to identify and characterize these cells from circulation and in tumor biopsies [5,6,7,8,18,23,24,25,26,27,28,29,30,31,32,33,34]. THCs were described in the circulation of patients with several different solid cancers, including colon, breast, ovarian, pancreatic, renal, esophageal, and melanoma [1,2,4,5,6,7,8,10,12,13,14,15,19,20,22,30,31,33,34]. Recently, we identified polyploid giant cells expressing CD61, Syncytin-1, telomerase, Runx2, and macrophage markers (CD11b, CD68, and CD163) in the circulation of patients with myelodysplastic syndromes (MDS) [35].

Recent advances in single-cell technologies and their application in characterizing tumors and their microenvironment (TME) has resulted in an explosion of datasets from various cancers that provide an opportunity for the identification and characterization of THC using these datasets, but there remain significant challenges in identifying and characterizing these cells. Analysis of scRNAseq data provides comprehensive information on THC, including their cellular origin, frequency, immune partner, gene expression profiles, and an opportunity to correlate their presence with clinical data and outcome. In this study, using scRNAseq and spatial datasets from colon and breast cancer and normal tissues, we provide evidence for the presence of HCs and comprehensively characterize their transcriptome.

## 2. Material and Methods

### 2.1. scRNAseq and Spatial Datasets

To identify a dataset for this analysis, we systematically searched publicly available cancer datasets. Given the rarity of hybrid cells, we focused on studies with the largest number of cells; hence, studies with less than 100,000 cells were removed. We reviewed the original publication for availability of clinical data. Other parameters include quality of single-cell data, cell-type annotations, and completeness of publicly available information.

The colorectal cancer dataset [36] was downloaded from https://www.ncbi.nlm.nih.gov/geo/query/acc.cgi?acc=GSE178341, and the breast cancer dataset [37] was downloaded from https://singlecell.broadinstitute.org/single_cell/study/SCP1039/a-single-cell-and-spatially-resolved-atlas-of-human-breast-cancers#study-download accessed on 16 January 2024.

The colorectal dataset, initially containing 371,223 cells, was filtered following quality control procedures described in Pelka et al. (2022) [36] to retain 370,115 cells. The breast cancer dataset consisted of 100,064 cells with provided barcodes, features, and matrix files. Both datasets included metadata for cell lineage and type.

The colorectal cancer and non-diseased colon Xenium spatial datasets were downloaded from https://www.10xgenomics.com/datasets/human-colon-preview-data-xenium-human-colon-gene-expression-panel-1-standard accessed on 30 January 2024. These datasets were generated using the pre-designed panel of 325 genes and the add-on panel of 100 additional genes. The colon cancer dataset was generated on FFPE-preserved tissue obtained from a stage 2A adenocarcinoma.

The Xenium breast cancer dataset was downloaded from https://www.10xgenomics.com/datasets/xenium-ffpe-human-breast-with-custom-add-on-panel-1-standard accessed on 27 February 2024. This dataset was generated using the pre-designed panel plus an add-on panel of 100 custom genes on a 5 μm section from an infiltrating ductal carcinoma (IDC) tumor of a breast cancer patient.

### 2.2. scRNAseq and Spatial Data Processing

Both the single-cell RNA-seq and spatial in situ datasets were analyzed using Seurat v5.0.1 [38]. A schematic overview of the analyses is shown in Appendix A. For each dataset, a Seurat object was created and processed. Expression data were normalized and scaled using SCTransform (v1.6.0), which also identified highly variable features. Principal component analysis (PCA) was used to reduce dimensionality, and the top 10 principal components were fed into a graph-based clustering algorithm with a resolution of 0.8. For visualization, UMAP was used to project the data down to two dimensions. The scCustomize package (v2.0.1) [39] and Seurat’s built-in visualization tools facilitated data exploration through dimplots, ImageDimPlots, violin plots, and dot plots. Additionally, for spatial data, we exported the cluster and group information to annotate the data visualized in Xenium explorer (Version 1.3, 10X genomics).

### 2.3. Doublet-Scoring Analysis

To identify and remove doublets, we leveraged the scds package (v1.18.0) [40]. First, the Seurat object was converted to a SingleCellExperiment format using as .SingleCellExperiment. We then applied three doublet-scoring methods: co-expression-based (cxds), binary classification-based (bcds), and a hybrid combining both approaches. Benchmarking revealed the hybrid approach outperformed cxds and bcds individually, so its score was added to the Seurat object for subsequent doublet filtering.

### 2.4. Hybrid Cell Identification

To mark hybrid cells, we used WhichCells function to mark cells if the expression *EPCAM* > 0, *KRT8* > 0, *CD14* > 0, and *CD163* > 0. Additionally, the cell’s doublet score should be less than or equal 0.5 (mean + 2 S.D.).

### 2.5. Differential Gene Expression

To identify cluster-specific marker genes, we employed differential expression analysis. We used FindAllMarkers() to compare each cluster against all others, and for more focused comparisons, the FindMarkers() function was used for individual pairs of clusters or groups. Both functions were called with default parameters. For each identified marker gene, we calculated several key metrics: the *p*-value for significance, the average fold change in expression between the cluster and the rest (avg_logFC), the percentage of cells expressing the gene within the specific cluster (pct.1), the percentage expressing it in all other clusters (pct.2), and the adjusted *p*-value (*p*_val_adj). Among these, avg_logFC was chosen as the primary indicator of cluster specificity.

To visualize the differentially expressed genes, we generated volcano plots using the VolcaNoseR shiny app (https://github.com/JoachimGoedhart/VolcaNoseR, accessed on 30 January 2024). Before uploading the results of the FindMarkers function to VolcaNoseR, we added a column containing the negative log10 of the *p*-values.

### 2.6. Functional Gene Set Enrichment Analysis

Gene set enrichment analysis was performed using the fgsea library in R. Genes were ranked according to their average log2fold change, and this ranking was used as the input to fgsea. In total, 10,000 gene permutations were used to calculate statistical significance, and a false discovery corrected *p*-value of 0.05 was required for statistical significance of a gene set. We used the hallmark gene sets available from the Human MSigDB collection (https://www.gsea-msigdb.org/gsea/msigdb/human/collections.jsp accessed on 21 January 2024).

### 2.7. Statistical Analysis

Statistical analyses were performed using GraphPad Prism (v10.1.2, GraphPad Software). Student’s *t*-test or Mann–Whitney test, as applicable, was used for statistical analysis, and *p*-values  <  0.05 were considered statistically significant. Overall survival (OS) analysis was performed using the Kaplan–Meier method. OS was calculated from the date of diagnosis to the date of last assessment or death and censoring data at the time patients were last known to be alive. Survival curves were compared using the log-rank test.

## 3. Results

### 3.1. Hybrid Cells in Colon Cancer and Normal Biopsies Are More Frequently Found in Myeloid Cell Clusters

To identify hybrid cells (HCs) in the tumor biopsies, we analyzed previously published single-cell sequencing data of 370,115 high-quality cells from tumor (257,251 cells) and adjacent normal (112,861 cells) tissue from patients with colon cancer [36]. This dataset is obtained by sequencing single cells from 64 tumors from 62 patients and adjacent normal tissue from a subset of 36 patients. A Louvain algorithm utilizing K-nearest neighbor (KNN) graph-based distance data was used to cluster the cells, resulting in 43 clusters which were then visualized and explored using the non-linear dimensional reduction technique uniform manifold approximation and projection (UMAP) (Appendix A).

To assign cell-type identity to clusters, we used the cell-type annotation provided in the original study as it was based on a robust two-step graph-clustering and non-negative matrix factorization (NMF) approach. Three levels of cell annotation were provided— clTopLevel, clMidwayPr, and cl295v11SubFull—each with a higher degree of sub-classification within the preceding level. The top-level annotation divided cells into seven major cell types, including myeloid, epithelial, T/natural killer [NK]/innate lymphoid cell [ILC] (TNKILC), plasma, B, mast, and stromal cells (Appendix A).

To identify HCs with both epithelial and macrophage characteristics, we marked cells that express RNA for both the epithelial marker (*EPCAM*) and macrophage marker (*CD163*). We identified 2126 (0.57%) HCs (Figure 1A). Alternatively, we also used a stringent criterion and marked HCs based on the presence of multiple tumor (*EPCAM* and *KRT8*) and macrophage (*CD163* and *CD14*) markers and identified 778 (0.21%) HCs (Figure 1B). In both the cases, HCs were predominantly found within clusters 9, 12, 14, and 21 corresponding to myeloid cell types (Figure 1A,B and Appendix A). We also found a small percentage of HCs in the same cluster as epithelial cells (Figure 1B).

### 3.2. Doublets Analysis Further Refines Hybrid Cells’ Identity

To identify doublets that are artifacts of droplet-based sequencing, we calculated doublet scores for all the top-level clusters and removed cells with a high doublet score of above 0.5 (mean + 2 S.D.) (Figure 1C). As expected, the median doublet score for HCs (0.29) was slightly higher compared to the rest of the top clusters but close to myeloid (0.27) and mast cell (0.28) clusters. After removing doublets, only 360,975 of the total 370,115 cells remained for further analysis. This also resulted in HC numbers reducing to 610 from 778.

For the rest of the analysis below, we defined HCs as *EPCAM+, KRT8+, CD163 +, and CD14+* with a doublet score of <0.5. Using this definition, greater than 96% of HCs were found with myeloid cells, followed by around 4% with epithelial cell types. The HC represents 1.5 and 0.01 percent of myeloid and epithelial cells, respectively. In addition to *EPCAM*, *KRT8*, *CD14*, and *CD163*, an HC expresses RNA for other epithelial markers, including cytokeratins *KRT9* (CK9) and *KRT19* (CK19), blood lineage marker *PTPRC* (CD45), and monocyte/macrophage marker *CD68* (Figure 1D).

### 3.3. Hybrid Cells in Colon Cancer and Normal Biopsies Are More Frequently Found with Macrophage/Monocyte Clusters

Since most of the HCs were found in the myeloid cluster, we next focused our attention on this group of cells to identify the exact identity of the myeloid cell types that HCs clustered with. For this, we used ‘clMidWayPr’ and ‘cl295v11subFull’ annotations provided by Pelka et al. [36]. The mid-level annotation identified 19 cell types (Appendix A), and the full annotation divided cells into 87 cell types (Appendix A). Unsupervised clustering of 19 cell types, identified within the seven top-level cell types, using these nine markers, clustered hybrids with monocyte/macrophage clusters (Figure 1E).

We extracted cells from the myeloid cluster of the tumor fraction; these cells were clustered in 10 myeloid cell types including, monocyte, macrophage-like, dendritic cells (DC) 1, DC2, DC2 C1Q+, DC IL22RA2, pDC, AS-DC, mregDC, and granulocyte (Appendix A). The HCs were clustered with monocytes and macrophages, but predominantly with macrophages (Figure 1F), with expression of *CD14* and *CD163* very similar to macrophages (Figure 1G).

### 3.4. Differential Gene Expression Analysis Revealed an Extensive Hybrid Transcriptome

To identify the extent of transcriptome remodeling in HCs from tumors (THC), we extracted the tumor fraction and compared the gene expression profile of THC with tumor epithelial cells (*EPCAM*+ and *KRT8*+) (Figure 1H and Appendix A) and THCs with tumor-associated macrophage/monocyte cells (*CD14*+ and *CD163*+) (Figure 1I and Appendix A). We found 3824 genes significantly (*p*_adj_ < 0.05) differentially expressed between THCs and epithelial cells. Also, we found 5800 genes significantly (*p*_adj_ < 0.05) differentially expressed between THCs and macrophage/monocyte cells (Figure 1H,I). Compared to the fusion partners, the percentage and/or expression of the top five genes from each comparison was relatively low in the THCs (Figure 1J).

### 3.5. The Majority of the Hybrid Cells Were Found in Tumor Tissue Compared to Normal

To identify the proportion of HCs within normal (NHC) and tumor tissue, we split the dataset into normal and tumor fractions. The percentage of HCs within the tumor fraction was 0.21, which was 4.2 times more than within normal tissue (0.05) (Figure 2A). It is important to note that the myeloid cell infiltration was also more in tumor tissue compared to normal (Figure 2B,C). We found that myeloid cells’ contribution was 14.8 percent of all cells within tumor tissue compared to only 2.4% in normal tissues (Figure 2B), but despite that, of the 610 HCs, 549 (90%) were from the tumor sections compared to only 61 (10%) from normal tissues (Figure 2D).

HCs were found in 60 of the 62 patient samples (Appendix A). Their number varied between patients, with some patients showing up to 1.8% of their tumor cells represented by HCs (Appendix A). As noted above, in 36 patients, there were cells derived from both tumor and adjacent normal tissue, which allowed us to compare the proportion of HCs within the matching normal and tumor tissue in each patient. Except for one patient, in whom we did not find any HCs in both normal and tumor tissue, in the rest of the 35 patients, almost half had no HCs in normal tissue (Figure 2E). In patients where HCs were present in both normal and tumor tissues, HCs were consistently more in tumors compared to normal tissue (Figure 2E).

### 3.6. THC Shows a Distinct Gene Expression Profile

Differential gene expression analysis between NHCs and THCs identified 33 genes that were significantly differentially (*p*_adj_ < 0.05) expressed (Appendix A, Figure 2F,G). The percentage of HCs and fold change varied for each gene. *EPCAM*, *KRT8*, and *KRT18* expressions were lower but not significantly different in normal cells, whereas *CD14* and *CD163* showed similar expressions (Figure 2H). *CD44* was highly expressed in THCs (Appendix A), whereas *SPP1* was uniquely present in THCs and not in NHCs (Appendix A).

Gene set enrichment analysis using hallmark pathways identified 19 pathways significantly (*p*_adj_ < 0.05) enriched in THCs compared to NHCs (Figure 2I and Appendix A). There was upregulation of inflammatory pathways and other notable pathways, including upregulation of KRAS signaling and EMT pathways (Figure 2J), which were among the top 10 enriched pathways (Appendix A).

### 3.7. THC Number within the Tumor and Their Relation to Clinical Features

We next focused our study on THCs. For this, we subset the tumor cells by subtracting all the cells including the HCs found in normal tissue. THCs were found in the tumors of all but two patients, with their percentage ranging from 0.02 to 1.8% of all tumor cells. The absolute numbers range from one to eighty-three with a median of six cells.

We next investigated the relationship of THCs with tumor, node, and metastasis (TNM) staging system, histology grade, and outcome including survival (Appendix A). We noted an increase in the percentage of THCs with the T stage, with higher stages having twice as many THCs compared to early stages (Figure 2K,L). The median number of THCs was higher in higher T stages, but these differences were not statistically significant. No significant differences were observed with node status in this dataset (Appendix A). The median number of THCs was slightly higher in stages and high histologic grade, but these differences were not statistically significant (Appendix A). No significant differences in percentage of THCs were observed between patients who entered metastasis compared to those who did not in this dataset (Appendix A). Survival was not significantly different between patients with a percentage of THC below or equal to the median (0.17) and above the median (Appendix A).

### 3.8. In Situ High-Resolution Spatial Mapping Confirms the Presence of THCs in Tumor Sections

To confirm the presence of HCs in tissue sections, we analyzed publicly available datasets of normal colon and tumor (stage 2A colon adenocarcinoma) sections that were analyzed using a recently developed Xenium high-resolution in situ targeted panel of 425 genes including *EPCAM*, *KRT8*, *CD14,* and *C163*. A KNN-graph-based clustering identified 30 and 37 clusters, respectively, in normal and tumor tissues (Appendix A). To identify HCs, we defined HCs as cells that co-expressed all four markers as we defined in the scRNAseq analysis. We identified 2026 (0.73%) and 6958 (1.19%) HCs in normal (Figure 3A,B) and tumor sections, respectively (Figure 3C,D), a much larger proportion compared to the scRNAseq (Figure 2B).

Similar to scRNAseq data, the gene expression profile of THCs was different from NHCs (Figure 3E), with at least three markers low in normal compared to the tumor cells (Figure 3F). The majority of genes that were either low or high in tumors from scRNAseq were also similarly low or high, respectively, in spatial tumor cells (Appendix A). *EPCAM* and *KRT8* were abundantly expressed in tumor epithelial tissue with scattered expression of *CD14* and *CD163* in monocytes/macrophages (Figure 3G). The HCs were scattered throughout the tumor tissue (Figure 3D and Appendix A).

### 3.9. THCs Are Found in scRNAseq and In Situ Spatial Data from Breast Cancer

To identify THCs in the tumor biopsies of other cancers, we analyzed previously published scRNAseq data of 100,064 cells from tumor tissue of patients with breast cancer [37]. This dataset is obtained by sequencing single cells from 26 patients. We identified 41 clusters that were distributed in nine major cell types (Appendix A) and 29 minor cell types (Figure 4A), including a myeloid cluster with four myeloid cell types including monocytes and macrophages (Figure 4B).

Using the definition as set above, that is expression of *EPCAM*, *KRT8*, *CD14*, and *CD163* and doublet score of less than 0.5, we identified 93 THCs (Figure 4B). Similar to colon cancer, THCs were predominantly found within the myeloid cell types, more closely to monocytes and macrophages (Figure 4B). In addition to *EPCAM*, *KRT8*, *CD14*, and *CD163*, THCs express RNA for other epithelial markers including cytokeratins *KRT9* (CK9) and *KRT19* (CK19), blood lineage marker *PTPRC* (CD45), and monocyte/macrophage marker *CD68* (Figure 4C).

We further confirmed the presence of THCs in tissue sections of breast cancer analyzed using a Xenium in situ targeted panel of 380 genes. A KNN-graph-based clustering identified 29 clusters (Appendix A). We identified 967 (0.16%) THCs in this tumor section (Figure 4D,E and Appendix A).

## 4. Discussion

In this study, we used publicly available datasets generated using two independent techniques from multiple cancers to identify hybrid cells with characteristics of CHCs using single-cell and spatial transcriptomic datasets. We found HCs both in normal and tumor tissues of cancer patient, but a much higher number in the tumor area. This finding is not surprising because cell fusions have been described in normal tissues and particularly bone marrow-derived cells are known to fuse with damaged or corrupted cells [41]. The critical finding here is that the transcriptomes of the two differ markedly with THCs showing higher KRAS and P53 signaling likely co-opted from tumor components (Appendix A). *TP53* and *KRAS* mutations are common in colorectal cancer patients [42]. THCs also show a higher EMT gene expression compatible with the notion that fusion with a macrophage allows tumor cells to escape their tissue of origin. *CD44* expression was higher in THCs compared to NHCs (Appendix A), consistent with previous observations [20], while *SPP1* was expressed exclusively in THCs with a subset having very high expression (Appendix A). *SPP1*+ macrophages have been implicated in the progression of and poor outcomes for colorectal cancer patients [43]. CD44 and SPP1 could be potential targets for eliminating THCs.

NHCs showing a hybrid expression profile could represent normal macrophages engulfing stressed cells as a part of normal cellular homeostasis. Such cells would show a hybrid status by expression profile but not by immunohistochemistry since there is no genomic fusion and reorganization such as that seen in THCs. We think this is likely the case with the HCs we are detecting in adjacent normal tissue showing a non-inflammatory transcriptome. We and others have previously proposed that fusion and/or phagocytic events within normal tissue could be an early “seed” of cancer origin and not just an early seed of metastasis [44,45,46,47]. A way to make this distinction between an NHC and a true hybrid cancerous “first cell” would be to show the two distinct lineage markers on the surface (by immunohistochemistry) along with a distinct inflammatory transcriptome (by scRNASeq).

While we clearly demonstrated the presence of THCs in tumors, the mechanism of origin remains unclear, with the dominant hypothesis being that the cause is cell fusion [9,15,47,48,49,50,51,52,53,54,55,56,57,58]. Previous studies suggested the role of syncytins in the fusion of tumor and immune cells. In this dataset, we observed a small fraction of macrophages/monocytes as well as epithelial cells expressing both syncytins, with the number of cells expressing RNA for syncytin-1 (*ERVW-1*) higher compared to syncytin-2 (Appendix A). Syncytin-2 (*ERVFRD-1*) expression was predominantly localized to mast cells (Appendix A). Within the myeloid component, syncytin-1 expression was localized to monocytes and macrophages. Syncytin-1 expression was not observed in hybrid cells (Appendix A). Syncytin-1 positive cells were observed in 44 out of 62 patients, even though THCs were present in patients where we did not find any syncytin-1 positive cells. Interestingly, the median number of hybrid cells was more in patients with syncytin-1 positive cells, but these results were not statistically significant (Appendix A). We speculate that the syncytin expression is required for fusion but either becomes diluted or is too low to be detected using scRNAseq, which is known to be biased towards high-expressing genes or the expression is turned off once the fusion process is complete and hence not detectable in hybrids. Future studies using targeted in situ measurements for syncytins’ RNA and/or protein and subsequent isolation of THCs for downstream studies will provide a clear role for syncytins in the hybrid cell formation.

It is possible that fusion events are higher in monocytes compared to macrophages, which immediately suggests that the origin of a macrophage-tissue cell hybrid may be a result of phagocytosis and monocyte-tissue cell hybrid the result of fusion. A recent study showed that tumor cells with low CD47 expression induce delayed phagocytosis resulting in incomplete digestion and subsequent THC formation [59]. As noted above, in our analysis comparing THCs with NHCs in the colon cancer dataset, we found increased KRAS and TP53 signaling in THCs, which, as we previously noted, are frequently deregulated in colon cancers.

It is likely that tumor cells fuse with each other and with cell types other than those of monocyte/macrophage lineage, but their presence in circulation is not well documented; hence, such a fusion may not provide an escape mechanism from tumor tissues but likely provides resistance to therapies by acquiring polyploidy [60,61,62,63].

There are multiple challenges in defining, identifying, and transcriptionally characterizing HCs from high-throughput datasets [64]. The first challenge was in defining the HCs and we used a stringent criterion to mark HCs. We marked cells as HC only when we detected (at any expression level) the presence of all four transcripts from genes *EPCAM*, *KRT8*, *CD14*, and *CD163*. CK8 is highly expressed in epithelial cells, including solid tumor cells and CD163 in macrophage lineages. We also included EPCAM and CD14, which also mark epithelial and monocyte/macrophage lineages, respectively. We selected these markers to be consistent with the markers used to characterize CHCs, but it is likely that other markers, including FCER1G, TYROBP, and AIF1 that are abundantly expressed in macrophage/monocyte lineages, can also be used to mark these cells. The hybrid nature of the HCs identified using the above four markers was further strengthened by the fact that these cells also showed expression of other epithelial (CK18, CK19), macrophage (CD68), and panleukocyte (CD45) markers (Figure 1E) even though they were not used to mark these cells. We further show that the gene expression pattern of THCs was different from epithelial and macrophage/monocyte cells with more than a thousand genes differentially expressed, suggesting the occurrence of the four markers used to mark these cells is not a stochastic process but involves significant transcriptome remodeling.

The second challenge was to distinguish HCs from doublets that are frequently observed in scRNAseq datasets. There are several different algorithms developed to remove these cells from further analysis to allow “single-”cell analysis. We reasoned that if the HCs were doublets of monocyte/macrophage and epithelial cells co-segregating together during droplet formation, then they would have either formed a distinct cluster due to doubling of the transcriptome, or if they did not form a distinct cluster, they would have had the same likelihood of being in an epithelial and myeloid cluster; thus, half of the HCs should have clustered with a monocyte/macrophage cluster and the other half with an epithelial cluster, but the majority of the HCs we found did not form a distinct cluster but were spread across myeloid and epithelial cells, with >75% in a myeloid rather than an epithelial cluster (Figure 1F). Within the myeloid cluster, the HCs clustered with macrophages rather than monocytes even though we used CD14, which predominantly marks monocytes. Also, we did not find a high doublet score for HCs, and the doublet score was similar to the myeloid cell cluster (Figure 1C). However, we did find a bimodal distribution in the HC score, with a small fraction of cells with high doublet scores likely representing true doublets. We used highly stringent criteria to remove doublets: a score of above 0.5 (mean + 2 S.D.). A typical droplet scRNAseq has a 0.1% doublet rate, but the criteria we used removed 2.5% of all cells from our analysis, including 21% of the HCs detected, without filtering for doublets. Thus, we are confident that the HCs identified are true hybrids and not artifacts. Further, using the Xenium in situ spatial dataset, we could show that the signal for the four makers was limited to single cells. We acknowledge that the percentage of HCs identified using Xenium in situ were more than scRNAseq. It is likely that we are underestimating the number of HCs in scRNAseq data due to the low depth of sequencing, which is typical of most scRNAseq platforms, which will likely eliminate cells that express low levels of any one of the four markers we used to define HC, but also because the sensitivity of in situ spatial methods are high. Also, as noted above, we used a rigid criterion for marking these cells when all four markers are present, but it is likely that cells that lack expression of any one of these four markers are not marked as hybrid. As discussed above, the use of doublet estimating methods might overcompensate and likely eliminate true HCs. Indeed, our HC estimates were reduced from 710 to 668. The use of doublet detection and elimination of doublet cells is a double-edged sword for HC detection in scRNAseq; while not removing them may result in true doublets contaminating the analysis, removing them may also remove some of the true hybrids because HCs by definition have a transcriptome of cells of two different lineages, and the majority of doublet detection programs rely on this fact. The predominant mechanism of HC formation is believed to be due to the fusion of monocytes/macrophages with cancer epithelial cells or via a “failed” phagocytosis attempt; thus, removing cells by scoring for doublets will remove these cells from the analysis. We suggest that for HC analysis, doublet scoring should be used cautiously.

Several high-plex in situ technologies, including Xenium In Situ (10X Genomics), CosMx (NanoString), MERSCOPE (Vizgen), and Molecular Cartography (Resolve), that offer single-cell resolution to spatial methods are currently in rapid development, and several of these have recently been commercialized. The major advantage of in situ spatial technology is its ability to measure gene expression on formalin-fixed paraffin-embedded (FFPE) tissues, such as the one used in this study, which will enable studies on archived tumor biopsies. Archived specimens in the repositories generally have longer follow-up data on patients, which will allow one to correlate HC presence and number with outcomes such as progression and survival. Additionally, these methods provide subcellular resolution, which may help to document the fusion process, including early and late stages during fusion by tracking transcripts specific to each parental cell type. Finally, in situ platforms are more sensitive; thus, low-expressed and rare transcripts can be identified and measured. We used a dataset that was generated using a preliminary version of the Xenium Human Colon Gene Expression Panel, but a final version may have more than 425 genes, increasing the number of genes that can be simultaneously analyzed. Current in situ panels are designed to study tumor immune microenvironments and have a limited number of genes involved in cell–cell communication, but these platforms can be customized to add additional genes. Hence, designing a panel focused on HC detection results in an explosion of data at single-cell resolution in space that will enable a comprehensive HCs characterization, including their location within tumors with respect to tumor geography, cell–cell interaction, communication, etc.

We also acknowledge the limitations of this study and its approach. The scRNAseq and spatial data are not derived from the same patients; thus, we cannot rule out that the difference in HCs observed using these two different methods could be just the differences in the HC number between patients, as we noted (Figure 2E and Appendix A) above that HC numbers vary between patients. The in situ spatial technology, including the software used to analyze the data, is under rapid development, and it is likely that a new user-friendly analysis pipeline and improved methods will help in the wider adoption of these technologies, which will further refine HC detection. These improved methods will provide further insight into their formation, immune evasion, tumor promotion, drug resistance, and metastasis. We did not observe any significant relationship between hybrid cells and survival or metastasis due to the short median follow-up of patients in the colon cancer dataset, which was only 2 years, and except for three patients, none of the patients developed metastases. As noted above, analysis of large retrospective datasets will enable studies assessing the relationship between outcomes and THCs in biopsies. Previous studies have shown a significant relationship between CHCs and survival in both colon and breast cancer patients [19]. The presence of macrophage traits in tumor biopsies has been shown to be associated with survival in both colon and breast cancer patients [23,24,25,28]. To allow for tumor purity and safety of the patients, the tumor tissue was not sampled from invasive borders, thus precluding the analysis of hybrid cells from these regions. We hope to address these relationships with much larger datasets with greater clinical details.

Several different technologies have been developed for the isolation of CHCs but isolating HCs from tumors remains challenging. We briefly discuss a few technologies and challenges for isolating THCs: (1) access to fresh or viable frozen biopsies: unlike CHCs that are readily available in peripheral blood which can be obtained through minimally invasive methods, THCs require tumor biopsies obtained through more invasive procedures. These biopsies need immediate processing or viable freezing for later use. (2) Single-cell isolation: CHCs and circulating tumor cells (CTCs) are single or small clusters suspended in blood, facilitating easy isolation. However, THCs reside within tissues, necessitating gentle methods for single-cell isolation [36]. Recent advancements in single-cell technologies provide methods for gentle tissue dissociation to yield single cells. (3) Distinguishing THCs from epithelial cells: distinguishing CHCs/CTCs from blood cells is relatively straightforward, and well-validated technologies exist for their isolation. In contrast, differentiating THCs from epithelial cells within tissues poses a challenge. We propose using multiplex immunofluorescence antibody panels targeting the specific markers we identified in our study to mark THCs for further isolation using flow or image cytometry. (4) Laser capture microdissection: tumor biopsy sections on slides can be used for THC isolation. Staining these sections with immunofluorescence antibody panels and methods that preserve cell viability allows for subsequent THC isolation using laser capture microdissection.

Here, we describe a framework for identifying hybrid cells using high-throughput scRNAseq and in situ spatial data from cancer and normal tissues, using colon cancer as a model, but we show that hybrid cells are present in other cancers. Ultimately, we hope to simultaneously characterize THCs from the tumors and circulation of the same patient to gain a better understanding of their role in tumor biology and to develop effective methods for targeting THCs. Since THCs are characterized by the expression of proteins from two different lineages that are infrequently found together on normal cells, a Boolean logic gate system (AND) can be implemented to specifically target the THC. For example, one protein specific from macrophage lineage and others specific to epithelial lineage can be selected for combinatorial targeting. Several approaches were described that use AND logic gates to target cell surface proteins, and these include but are not limited to CoLOCKR [65], SynNotch CAR [66,67], Split-CAR [68], and SUPRA CAR [69]. Alternatively, intracellular inputs, such as transcription factors, can also be targeted [70].

## 5. Conclusions

This study has established a robust framework for identifying hybrid cells in tumor biopsies analyzed using high-throughput approaches. This approach provides compelling evidence for the presence and frequency of HCs that reside within tissues and offers a comprehensive characterization of their transcription activity and functional state. Importantly, the study pinpoints potential differences in HCs between normal and tumor cells and identifies pathways that may contribute to their increased dissemination throughout the body. These findings raise the exciting possibility of targeting HCs for novel cancer therapies. However, further research is crucial to fully elucidate the role of HCs in cancer progression and their potential contribution to therapy resistance. By delving deeper into the biology of these elusive cells, we can pave the way for more effective and personalized cancer treatments in the future.

## 6. Patents

A.M.A. and A.R. are co-inventors on a pending patent application describing the targeting of giant and/or hybrid cells.

## Figures and Tables

**Figure 1 cancers-16-01444-f001:**
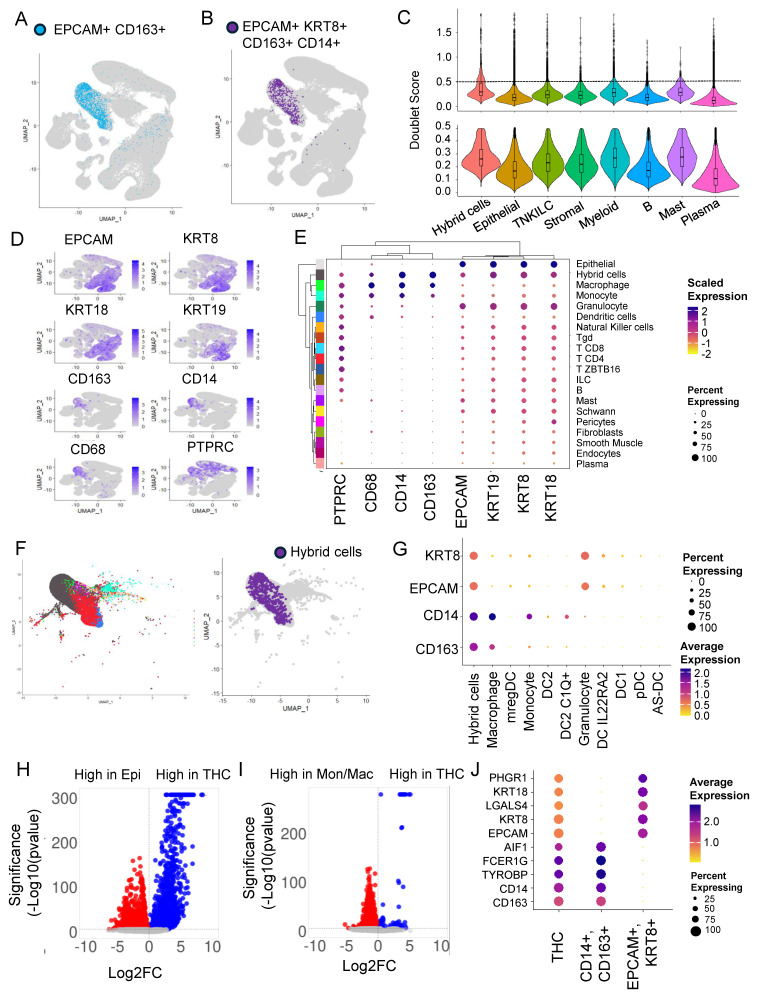
Hybrid cells are found in tumor and normal colon biopsies, frequently associated with myeloid clusters, and show distinct gene expression profiles compared to parental cells. (**A**) Un-form manifold approximation and projection (UMAP) plot of all cells from scRNAseq data. Hybrid cells are marked with only two markers (*EPCAM* and *CD163*) highlighted in blue and the rest of the cells in light grey. (**B**) UMAP plot of all cells with hybrid cells marked with four markers (*EPCAM*, *KRT8*, *CD14*, *CD163*) highlighted in purple and the rest in light grey. (**C**) Violin plots with box plots’ doublet score before (top) and after (bottom) filtering out for cells with >0.5 score. (**D**) UMAP plot showing expression of four marker genes (*EPCAM*, *KRT8*, *CD14*, *CD163*) and other epithelial (*KRT18* and *KRT19*) and monocyte/macrophage (*CD68*) and leukocyte (*CD45*) genes. (**E**) Clustered dot plot of the eight genes noted in D above and 19 cell types along with hybrid cells showing hybrid cells closely clustering with monocyte and macrophage cluster. (**F**) UMAP plot of myeloid subcluster showing 10 myeloid clusters (left) and hybrid cells (right). The hybrid cells are found in both monocyte and macrophage clusters but predominantly in macrophage. (**G**) Dot plot of myeloid subcluster. (**H**) Volcano plot of differential gene expression between epithelial cells (marked by expression of *EPCAM* and *KRT8*) and tumor hybrid cells. (**I**) Volcano plot of differential gene expression between monocyte/macrophage cells (marked by expression of *CD14* and *CD163*) and tumor hybrid cells. (**J**) Dot plot of top 5 genes that were differentially expressed between epithelial vs. tumor hybrid cells comparison and monocyte/macrophage vs. tumor hybrid cells.

**Figure 2 cancers-16-01444-f002:**
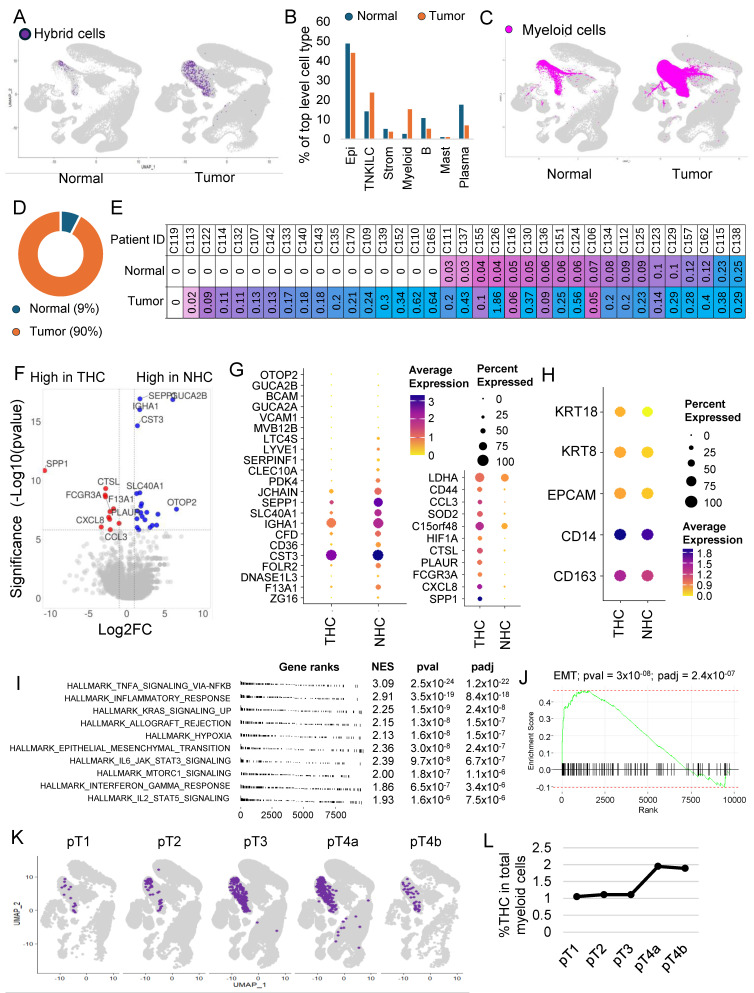
Hybrid cells are more frequent in colon tumors and show a distinct gene expression profile marked by upregulation of inflammatory pathways and epithelial–mesenchymal transition genes. (**A**) UMAP plot of all cells with hybrid cells highlighted in purple and the rest in light grey in normal and tumor tissue. (**B**) A bar plot showing the percentage of different cell types between tumor and normal tissues in the dataset. (**C**) UMAP plot of all cells with all myeloid cells highlighted in purple and the rest in light grey in normal and tumor tissue. (**D**) Donut plot showing the percentage of hybrid cells from each tissue type. (**E**) Table showing patient ID and percentage of hybrid cells in normal and tumor tissue of each patient where a matching normal tissue was available. (**F**) Volcano plot of differential gene expression between hybrid cells from tumor and normal tissues. (**G**) Dot plot of 33 top differentially expressed genes (*p*_adj_ < 0.05) between hybrid cells from tumor and normal tissues. (**H**) Dot plot showing expression of four marker genes and *KRT18* in hybrid cells from normal and tumor tissues. (**I**) A summary gene set enrichment analysis (GSEA) plot of top 10 hallmark pathways along with normalized enrichment scores (NES), *p*-value (*p*_val_), and p-adjusted (*p*_adj_) values. Sorted with the most significant pathways at the top. (**J**) Enrichment plot of epithelial–mesenchymal transition (EMT) pathway. (**K**) UMAP plot of all cells, with tumor hybrid cells highlighted in purple and the rest in light grey in cells subsetted based on the stage of the tumor. (**L**) Line plot showing the percentage of tumor hybrid cells within the myeloid cells in different stages of tumor.

**Figure 3 cancers-16-01444-f003:**
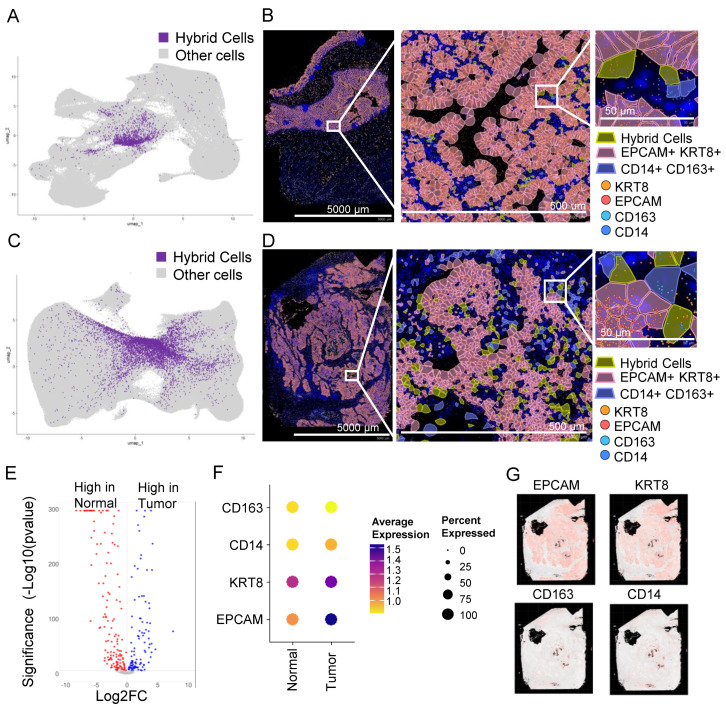
Hybrid cells are found in tumor and normal colon biopsy sections assayed using in situ high-resolution spatial mapping and show distinct gene expression profiles. (**A**) UMAP plot of all cells with hybrid cells highlighted in purple and the rest in light grey in a normal tissue section of a non-cancer donor analyzed using Xenium spatial seq. (**B**) Spatial plot of normal section showing hybrid cells (*EPCAM*+, *KRT8*+, *CD14*+, and *CD163*+), epithelial cells (*EPCAM*+ and *KRT8*+), and monocyte/macrophage (*CD14*+ and *CD163*+) cells in a section from a normal colon at 1× (left, entire section), 10× (middle; zoomed), and 100× (right; zoomed). The zoomed region was manually selected to show a region where all three types of cells were present. (**C**) UMAP plot of all cells with hybrid cells highlighted in purple and the rest in light grey in tumor tissue section of colon cancer patient analyzed using Xenium spatial seq. (**D**) Spatial plot of tumor section showing hybrid cells (*EPCAM*+, *KRT8*+, *CD14*+, and *CD163*+), epithelial cells (*EPCAM*+ and *KRT8*+), and monocyte/macrophage (*CD14*+ and *CD163*+) cells in a section from a colon tumor at 1× (left, entire section), 10× (middle; zoomed), and 100× (right; zoomed). The zoomed region was manually selected to show a region where all three types of cells were present. (**E**) Volcano plot of differential gene expression between hybrid cells from normal and tumor tissues shown in Figure 3A and Figure 3C, respectively. (**F**) Dot plot showing expression of four marker genes in hybrid cells from normal and tumor tissues. (**G**) Spatial plot of tumor section showing expression of four marker genes.

**Figure 4 cancers-16-01444-f004:**
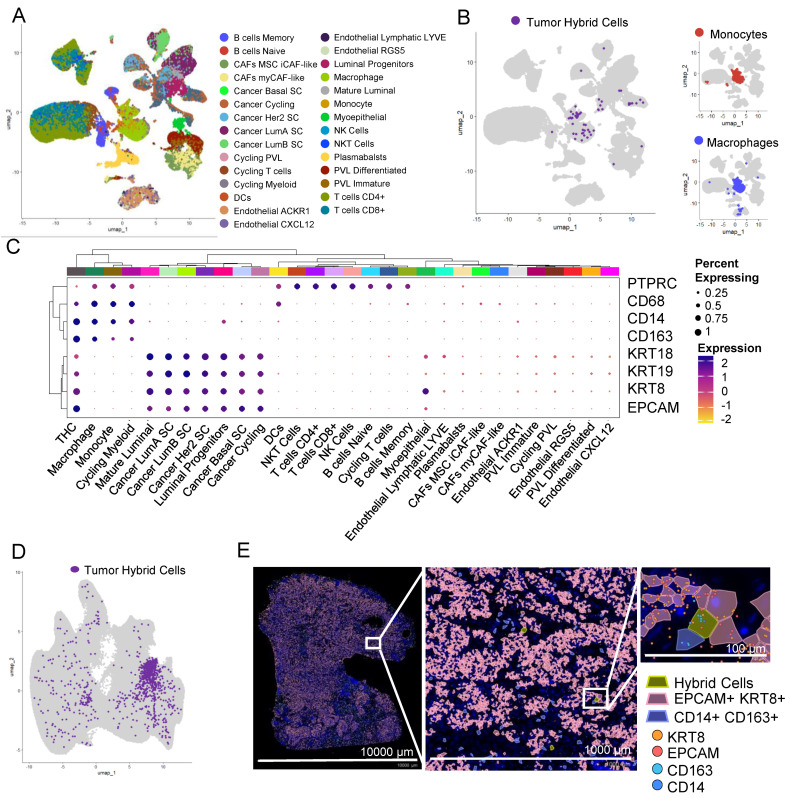
Hybrid cells are found in tumor single cells and biopsy sections from breast cancer. (**A**) UMAP plot of all cells in scRNAseq data from breast tumor tissue. Cells are colored by their cellular identity as reported in Wu et al. (**B**) UMAP plot of all cells with hybrid cells highlighted in purple and the rest of the cells in light grey in breast tumor tissue. The myeloid cluster is marked with a dashed line. On the right, the UMAP plot of all myeloid cells with monocytes (top) and macrophages (bottom) is highlighted in red and blue, respectively. The rest of the cells are highlighted in light grey. (**C**) Clustered dot plot of 30 cell types along with hybrid cells, showing hybrid cells closely clustering with monocyte and macrophage clusters. (**D**) UMAP plot of all cells with hybrid cells highlighted in purple and the rest in light grey in tumor tissue section of breast cancer patient analyzed using Xenium spatial seq. (**E**) Spatial plot of tumor section showing hybrid cells (*EPCAM*+, *KRT8*+, *CD14*+, and *CD163*+), epithelial cells (*EPCAM*+ and *KRT8*+), and monocyte/macrophage (*CD14*+ and *CD163*+) cells in a section from a breast tumor at 1× (left, entire section), 10× (middle; zoomed), and 100× (right; zoomed). The zoomed region was manually selected to show a region where all three types of cells were present.

## Data Availability

All the data are publicly available. The colorectal cancer dataset [35] was downloaded from https://www.ncbi.nlm.nih.gov/geo/query/acc.cgi?acc=GSE178341, and the breast cancer dataset [36] was downloaded from https://singlecell.broadinstitute.org/single_cell/study/SCP1039/a-single-cell-and-spatially-resolved-atlas-of-human-breast-cancers#study-download accessed on 16 January 2024. The colorectal cancer and non-diseased colon Xenium spatial datasets were downloaded from https://www.10xgenomics.com/datasets/human-colon-preview-data-xenium-human-colon-gene-expression-panel-1-standard accessed on 30 January 2024. The Xenium breast cancer dataset was downloaded from https://www.10xgenomics.com/datasets/xenium-ffpe-human-breast-with-custom-add-on-panel-1-standard accessed on 27 February 2024.

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
