# Peer review of "scRNAseq and High-Throughput Spatial Analysis of Tumor and Normal Microenvironment in Solid Tumors Reveal a Possible Origin of Circulating Tumor Hybrid Cells"

_cancers, 2024, doi:10.3390/cancers16071444_

Round 1

Reviewer 1 Report

Comments and Suggestions for Authors

This submission, entitled "scRNAseq and high-throughput spatial analysis of tumor and normal microenvironment in solid tumors reveal a possible origin of circulating tumor hybrid cells", describes a method for analyzing published cancer and normal cell single cell sequence datasets to identify gene expression profiles in hybrid cells (HC) in both normal and tumor tissue. They then used Xenium spatial in situ datasets of stage 2A adenocarcinoma colon cancer as well as a ductal carcinoma breast cancer to validate the results and localize "marked" cells.

Stringent selection criteria were used to eliminate doublets, using 3 scoring methods, acknowledging that this may have also excluded hybrid cells, which could look like doublets. The results suggest that, based on the authors' gene expression profile selection, hybrid cells are found in both normal and cancer tissue, but tend to be found in much greater numbers in cancer tissue. In addition, they found that hybrid cells almost exclusively cluster with monocyte/macrophage clusters rather than other cell types. They offer a possible explanation of the origin of these hybrids, with macrophage hybrids being the result of phagocytosis and monocyte hybrids to occur after fusion with tissue cells.

The authors describe various challenges, including defining HCs, and using high-throughput datasets to identify and transcriptionally characterize these cells. It is important to understand the advantages and pitfall of using these single cell genomic and transcriptome databases to identify these very low abundance cell types. 

Identifying the existence of these cells and the pathways that may promote tumor dissemination, since circulating HCs may be able to initiate distant metastatic lesions, will be an important advancement. Yet to come is exploration of their function, independent of their gene expression profile, and their roles in immune evasion, tumor promotion, drug resistance and metastasis.

So, while technology has advanced to the point where many single cells can be analyzed for very specific gene expression profiles, characterization of their phenotypes, when they reside in tissues or found in the circulation will be important in determining their impact on tumor progression and if studies designed to target these cells are warranted.

As part of Methods or Introduction, a clearer description of their overall data analysis process would help a reader who is not very familiar with the sorting/clustering software programs and the multiple steps involved in producing the plots. This would help readers see the  big picture before delving into the results.

Some of the figure text is hard to read and if this can not be improved during the publication process, they should be redone. SFig4, Fig 2 panels I and J.

Overall, this manuscript is well written, with methods clearly stated, although many readers would appreciate additional descriptions of some of the analytical software programs used, as there were quite a few of them. The authors provide clear explanations of their gene expression criteria and how the results of each analysis fits into their overall conclusions.

Comments on the Quality of English Language

English language is adequate, other than a few typos some minor inconsistencies, such as using artefact and artifact.

Author Response

  1. Summary

We thank this reviewer for their time to review this work and for providing an excellent summary of our work. We especially appreciate the comments of the reviewer on future directions for the field of hybrid cells. We agree with the reviewer that identifying the existence of hybrid cells in biopsies and the pathways that govern tumor dissemination leading to metastasis is just the first, but an important step that we describe in this manuscript, and we agree that exploring the roles of hybrid cells in immune evasion, tumor promotion, drug resistance and metastasis at functional level utilizing various in vitro and in vivo models is warranted. We believe that our study paves the way for this research.

Please find the detailed responses below and the corresponding revisions/corrections highlighted/in track changes in the re-submitted files.

  1. Point-by-point response to Comments and Suggestions for Authors

Comments 1: As part of Methods or Introduction, a clearer description of their overall data analysis process would help a reader who is not very familiar with the sorting/clustering software programs and the multiple steps involved in producing the plots. This would help readers see the  big picture before delving into the results.

Response 1: We have revised the methods and added a new supplemental figure 1 to present a schematic of overall data analysis. The changes are in page 3, lines 81 to 85, and line 108.

Comments 2: Some of the figure text is hard to read and if this can not be improved during the publication process, they should be redone. SFig4, Fig 2 panels I and J.

Response 2: We have revised the supplementary figure 4 (now supplementary figure 5), Fig 2 panels I and J, to improve the text.

Comments on the Quality of English Language

English language is adequate, other than a few typos some minor inconsistencies, such as using artefact and artifact.

Response 3: We have checked for typos and inconsistencies where applicable and revised the manuscript.

Reviewer 2 Report

Comments and Suggestions for Authors

This research paper sheds light on the potential origin of circulating tumor hybrid cells and their unique transcriptional profile. In current cancer research, Otto Aichel's theory on cell fusion has gained significance over time and the formation of hybrid cells between cancerous and healthy cells is recognized as a significant process relevant to the genesis of metastatic cancer cells. Tumor Hybrid Cells (THC) attract increasing attention and are seen as a promising avenue for new approaches to cancer treatment, offering insights into tumor heterogeneity, drug resistance, and metastatic processes. This asks for providing compelling evidence for the residence of hybrid cells in tumor tissues.

Overall, I found the paper well-written, documented, and constructed. The experimental steps are consistent and, involve a comprehensive approach to identify and characterize hybrid cells in solid tumors and normal healthy tissue using high-throughput single-cell and spatial transcriptomic techniques.

MAJOR ISSUE:

Despite the high-quality results obtained, a limitation of this study lies in its exclusive reliance on in silico/computational analysis as it lacks furnishing evidence of their verifiable existence.  Isolating tumor hybrid cells (THCs) for experimental validation and characterization can be a challenging but feasible task. I understand that this demonstration requires specific expertise besides a certain amount of time that probably goes beyond the paper’s objectives. However, the authors should at least discuss strategies and techniques that can be employed to isolate THCs and provide evidence of their effective existence. Indeed, the isolation of individual THCs or clusters of THCs for downstream analysis can be enabled by using antibodies specific for THC-associated markers with laser capture microdissection or cell sorting techniques (such as FACS and MACS).

MINOR ISSUES:

I have just some commentaries which I invite the authors to answer to:
1. The variation in the number of hybrid cells (HCs) among patients in a study raises the question of whether this variability could be influenced by the intrinsic rate of cell fusion in different individuals. It is plausible that differences in the percentage of HCs observed across patients may correlate with the expression of pro-fusion genes or factors that promote cell fusion.
Utilizing single-cell RNA sequencing data from patient samples, correlation analyses could be performed between the abundance of HCs in patient samples and the expression levels of candidate genes known to be involved in cell fusion processes. This analysis could help identify potential associations between fusion-related gene expression and the presence of HCs and eventually indicate if cancer-associated variations in their expression could account for the major propensity of tumor cells to form HCs.

2. The authors suggest that syncytins may be involved in HC formation, even though they are not detectable in THCs, possibly due to downregulation after cell fusion. Alternatively, it might be hypothesized that the different percentages of HCs among patients could correlate with the variability in the expression of syncytins in non-hybrid cells. It is conceivable that higher expression levels of syncytins in non-hybrid cells may indicate a greater propensity for cell fusion events, potentially leading to an increased formation of HCs within the tumor.

3. If variations in syncytin/pro-fusion genes expression levels indeed correlate with the propensity to form THCs, their levels could serve as predictive biomarkers for identifying patients who are more likely to develop THCs and stratifying patients based on their risk of THC formation and potentially guide personalized treatment strategies.

4. The discovery of hybrid cells in tumors could have significant implications for the development of new therapeutic targets for metastatic cancer treatment. However, targeting THCs for cancer therapy poses challenges due to their limited numbers and unique characteristics. It would be appreciable if the authors could add a brief discussion about strategies to overcome these challenges and effectively target THCs.

Author Response

  1. Summary

We thank this reviewer for their time to review this work and for providing their comments and suggestions. We are particularly pleased to know that the reviewer found the manuscript well-written and the approach consistent and comprehensive.

Please find the detailed responses below and the corresponding revisions/corrections highlighted/in track changes in the re-submitted files.

  1. Point-by-point response to Comments and Suggestions for Authors

MAJOR ISSUE:

Comments 1.  Despite the high-quality results obtained, a limitation of this study lies in its exclusive reliance on in silico/computational analysis as it lacks furnishing evidence of their verifiable existence.  Isolating tumor hybrid cells (THCs) for experimental validation and characterization can be a challenging but feasible task. I understand that this demonstration requires specific expertise besides a certain amount of time that probably goes beyond the paper’s objectives. However, the authors should at least discuss strategies and techniques that can be employed to isolate THCs and provide evidence of their effective existence. Indeed, the isolation of individual THCs or clusters of THCs for downstream analysis can be enabled by using antibodies specific for THC-associated markers with laser capture microdissection or cell sorting techniques (such as FACS and MACS).

Response 1: We agree with the reviewer that isolating THC from biopsies is crucial for not only confirming their presence but for further characterization. This study documented the presence and frequency of THC across different patients and cancers using single-cell sequencing data obtained from patient samples. We validated their presence spatially with an independent in situ method, again using patient samples. However, isolating and further characterizing THC falls outside the scope of this work as the objective was to provide evidence for their presence in tumor biopsy as a likely source of circulating hybrid cells (CHC) which are well characterized for their role in metastasis. Nevertheless, our study provides compelling evidence for the presence and frequency of THCs in biopsies, paving the way for future isolation studies.

As requested by the reviewer, we have now discussed the techniques and challenges associated with isolating THC and contrasting them with CHC, for which well-established isolation technologies exist. These changes now appear on page 11/12, lines 456-473.

MINOR ISSUES:

I have just some commentaries which I invite the authors to answer to:

Comments 1. The variation in the number of hybrid cells (HCs) among patients in a study raises the question of whether this variability could be influenced by the intrinsic rate of cell fusion in different individuals. It is plausible that differences in the percentage of HCs observed across patients may correlate with the expression of pro-fusion genes or factors that promote cell fusion. Utilizing single-cell RNA sequencing data from patient samples, correlation analyses could be performed between the abundance of HCs in patient samples and the expression levels of candidate genes known to be involved in cell fusion processes. This analysis could help identify potential associations between fusion-related gene expression and the presence of HCs and eventually indicate if cancer-associated variations in their expression could account for the major propensity of tumor cells to form HCs.

Response 1: Multiple theories are proposed to explain THC formation, with fusion promoted by fusogenic proteins such as syncytins being one such theory but evidence for delayed phagocytosis especially in the context of macrophage-mediated phagocytosis has also been provided. We discussed the most prominent theories. It is likely that the intrinsic rate of cell fusion could be different in different individuals regulated based on the expression of fusogenic proteins. We did a finer analysis and provided additional data in supplementary figure 10. We also modified the discussion to reflect this, the changes appear on page 9, lines 335 to 350.

Comments 2. The authors suggest that syncytins may be involved in HC formation, even though they are not detectable in THCs, possibly due to downregulation after cell fusion. Alternatively, it might be hypothesized that the different percentages of HCs among patients could correlate with the variability in the expression of syncytins in non-hybrid cells. It is conceivable that higher expression levels of syncytins in non-hybrid cells may indicate a greater propensity for cell fusion events, potentially leading to an increased formation of HCs within the tumor.

Response 2: As noted above, we provide additional data on syncytins in supplementary figure 10. We also modified the discussion to reflect this, the changes appear on page 9, lines 335 to 350.

Comments 3. If variations in syncytin/pro-fusion genes expression levels indeed correlate with the propensity to form THCs, their levels could serve as predictive biomarkers for identifying patients who are more likely to develop THCs and stratifying patients based on their risk of THC formation and potentially guide personalized treatment strategies.

Response 3: Syncytin-1 positive cells were observed in 44 out of 62 patients even though THCs were present in patients where we did not find any syncytin-1 positive cells. Interestingly, the median number of hybrid cells were more in patients with syncytin-1 positive cells, but these results were not statistically significant (Supplementary Figure S10E). The data presented neither rule out the role of syncytins nor provide conclusive evidence for their role. We hope to address this in future work supported by experimental data.

Comments 4. The discovery of hybrid cells in tumors could have significant implications for the development of new therapeutic targets for metastatic cancer treatment. However, targeting THCs for cancer therapy poses challenges due to their limited numbers and unique characteristics. It would be appreciable if the authors could add a brief discussion about strategies to overcome these challenges and effectively target THCs.

Response 4: We have modified the discussion to briefly discuss the approaches to effectively target THCs. These changes appear in the discussion, on page 12, lines 479-486.

Reviewer 3 Report

Comments and Suggestions for Authors

scRNAseq and high-throughput spatial analysis of tumor and normal microenvironment in solid tumors reveal a possible origin of circulating tumor hybrid cells

Ali et.al, current study utilizes publicly available datasets, employing both single-cell and spatial transcriptomic techniques, to identify and characterize hybrid cells (HCs) with characteristics of cancer hybrid cells (CHCs). The research spans various cancers, with a particular focus on colon cancer. The findings reveal the presence of HCs in both normal and tumor tissues, with a higher frequency observed in tumor areas. Notably, transcriptomic differences indicate the potential co-option of signaling pathways, such as KRAS and P53, from the tumor component. The study explores the implications of TP53 and KRAS mutations, common in colorectal cancer patients, and highlights elevated expression of EMT genes in HCs, suggesting a potential mechanism for tumor cells to escape their tissue of origin. The study also identifies specific markers like CD44 and SPP1 as potential therapeutic targets for eliminating HCs. Overall, the study contributes valuable insights into the field of hybrid cells in cancer. However, addressing the major and minor comments will enhance the clarity and completeness of the manuscript.

Major Comments:

1.      Clarify relationship between hybrid cells and patient outcomes by addressing the short median follow-up period and limited metastasis data in the colon cancer dataset.

2.      Author did not mention potential variation in HC numbers between patients observed in Figure 2E and S2 or it is not well clear.

3.      Author have not much detailed on the relationship between hybrid cells and survival or metastasis based on larger datasets with greater clinical details.

4.      Author could elaborate on the impact of in situ spatial technologies on HC detection and discuss potential improvements in future studies.

Minor Comments:

Proofread the manuscript for minor grammatical and typographical errors.

Ensure consistent formatting and citation styles throughout the manuscript.

Consider providing more context on the significance of specific markers used for HC identification.

Comments on the Quality of English Language

Proofread the manuscript for minor grammatical and typographical errors.

Ensure consistent formatting and citation styles throughout the manuscript.

Consider providing more context on the significance of specific markers used for HC identification.

Author Response

  1. Summary

We thank this reviewer for their time to review this work and for providing suggestions to improve the manuscript.

Please find the detailed responses below and the corresponding revisions/corrections highlighted/in track changes in the re-submitted files.

  1. Point-by-point response to Comments and Suggestions for Authors

 Major Comments:

Comments 1.  Clarify the relationship between hybrid cells and patient outcomes by addressing the short median follow-up period and limited metastasis data in the colon cancer dataset.

Response 1: We have now performed the analysis to clarify the relationship between hybrid cells and patient outcomes in the colon cancer dataset. The results are presented in a new supplemental figure 6. Additionally, we have now included a table (supplementary table 4) showing the percentage of THC in each patient and associated clinical data as reported in Pelka et al. We modified the main text to include the results of the analysis (page 7, lines 264-269). We also modified the discussion to reflect the new analysis (page 11, lines 421 to 427). We have modified the methods section to include the methodology used to analyze this data (page 4, lines 153-156).

Comments 2.  Author did not mention potential variation in HC numbers between patients observed in Figure 2E and S2 or it is not well clear.

Response 2: We noted the variation in HC numbers between patients in the results section on page 6, lines 230 to 238. We discussed these differences in the discussion section on page 11, lines 436 and 437. Additionally, we have now performed analysis by stratifying patients based on HC numbers and relating the number with survival. The results are presented in a new supplemental figure 5.

Comments 3. Author have not much detailed on the relationship between hybrid cells and survival or metastasis based on larger datasets with greater clinical details.

Response 3: We have reviewed several single cell datasets available at GEO/SRA, Single Cell Expression Atlas, Single Cell Portal, CZ Cell x Gene Discover, scRNAseq package on Bioconductor, and PanglaoDB for larger datasets with greater clinical details, but we could find any dataset much larger than the one we analyzed in this study and with such greater clinical details. If the reviewer knows any such dataset or has access to such data, we are happy to analyze. The colon cancer dataset we analyzed has > 350,000 cells from >60 patients with detailed clinical data but with limited follow-up data. Previous studies have documented a relationship between circulating hybrid cells and survival outcomes in colon cancer patients and we have reviewed this works in introduction and discussion. Additionally, we have reviewed the work in both introduction and discussion describing relationships between presence of macrophage-like characteristics in tumor biopsies and their relationship to survival and progression in both colon and breast cancer patients. This information appears on page 11, lines 444-447.

Comments 4.  Author could elaborate on the impact of in situ spatial technologies on HC detection and discuss potential improvements in future studies.

Response 4: We have now discussed the impact of in situ spatial technologies on HC detection and discuss potential improvements in future studies. This changes now appear on page 10, lines 419-427 and lines 430-433.

Minor Comments:

Comments 5. Proofread the manuscript for minor grammatical and typographical errors.

Response 5: We have reviewed the manuscript for grammatical and typographical errors and corrected them throughout the manuscript.

Comments 6. Ensure consistent formatting and citation styles throughout the manuscript.

Response 6: We have revised the citation style for consistency.

Comments 7. Consider providing more context on the significance of specific markers used for HC identification.

Response 7: We used these four markers because they are lineage-specific and has been previously used to characterize circulating hybrid cells. This information appears in the discussion section on page 9, lines 368-371.

Comments on the Quality of English Language

Comments 8. Proofread the manuscript for minor grammatical and typographical errors.

Response 8: We have reviewed the manuscript for grammatical and typographical errors and corrected them throughout the manuscript.

Round 2

Reviewer 3 Report

Comments and Suggestions for Authors
Authors made changes as per the suggestions. However, author should provide processing and descriptive R/python codes as well for the update version of manuscript to review and replicate at our end. Sometimes, methods are not very descriptive to understand in the article.

Author Response

Response to Reviewer 3 Comments

Point-by-point response to Comments and Suggestions for Authors

Comments 1: Authors made changes as per the suggestions. However, author should provide processing and descriptive R/python codes as well for the update version of manuscript to review and replicate at our end. Sometimes, methods are not very descriptive to understand in the article.

Response 1: Thank you for your quick review of the revised manuscript. We have added a supplementary file with all the code used to analyze the data and generate the figures. We have modified the Code Availability section to add the following: “The R script to reproduce the figures is provided as a supplementary file.”